# Microstructures and High Temperature Tensile Properties of As-Aged Mg-6Zn-1Mn-4Sn-(01, 0.5 and 1.0) Y Alloys

**Guangshan Hu [1,2], Meipeng Zhong [1,*] and Changfa Guo [1]**

[1] College of Mechanical and Electrical Engineering, Jiaxing University, Jiaxing 314001, China; huguangshan88@163.com (G.H.); changfa.guo@foxmail.com (C.G.)

[2] Zotye Automotive Engineering Research Institute, Hangzhou 310018, China

\* Correspondence: zhongmeipeng@mail.zjxu.edu.cn; Tel.: +86-188-5738-0257

**Abstract:** The microstructures and high-temperature tensile properties of as-aged Mg-6Zn-1Mn-4Sn-(0.1, 0.5 and 1.0) Y (wt.%, ZMT614-Y) alloys were investigated by optical microscopy (OM), X-ray diffractometer (XRD), scanning electron microscopy (SEM), transmission electron microscopy (TEM), and high-temperature tensile tests. The tensile temperatures were 150 °C, 200 °C, 250 °C and 300 °C, respectively. The results showed that the phase compositions of as-aged alloys were $\alpha$-Mg, $\alpha$-Mn, $MgZn_2$, $Mg_2Sn$, and MgSnY phases. The $Mg_2Sn$ and MgSnY high-temperature phases inhibited grain growth in the heat treatment and tensile processes. The as-aged ZMT614-0.5Y alloy has the best high-temperature mechanical properties, with yield strength (YS), ultimate tensile strength (UTS), and elongation values of 277 MPa, 305 MPa, and 16.7%, respectively, at 150 °C. As the tensile temperature increased to 300 °C, the YS and UTS decreased to 136 MPa and 150 MPa, and elongation increased to 25.5%. The fracture mechanism changed as the tensile temperatures ranged from 150 °C to 300 °C, from the transgranular fracture type at temperatures of 150 °C and 200 °C, to the transgranular and intergranular mixed-mode fracture type when tensile temperatures increased to 250 °C, to an intergranular fracture mechanism at 300 °C.

**Keywords:** Mg-6Zn-1Mn-4Sn-(0.1; 0.5 and 1.0) Y alloys; microstructures; high-temperature tensile properties; fracture mechanism

## 1. Introduction

Due to their low density and high specific strength, Magnesium (Mg) and its alloys have been widely used in the automotive and aerospace industries [1–3]. A Mg-Zn system, whose eutectic temperature is 340 °C and maximum solid solubility of Zn is 6.2 wt.%, dropping sharply at room temperature, has been recognized as a precipitation-hardenable system [4]. The main precipitate of Mg-Zn alloys is the $MgZn_2$ phase, which has a relatively low eutectic temperature [5]. When the deformation temperature is over 300 °C, the $MgZn_2$ phase becomes soft and acts as a lubricant, facilitating the grain boundaries sliding, and resulting in a significant decrease in strength [6]. Due to the limitations that the poor high-temperature performance of Mg alloys imposes on potential applications, the development of new heat-resistant Mg alloys with excellent mechanical properties has attracted considerable attention. For the formation of heat-resistant RE-containing phases (RE, rare earth), the addition of RE elements such as Nd, Ce, and La is a promising way to improve the high-temperature mechanical properties of Mg-Zn alloys in particular [7–10]. As one of the more cost-effective RE elements, yttrium (Y) has been widely used in Mg-Zn alloys [11–13]. Its maximum solid solubility is 12.4 wt.%, which falls to nearly zero at room temperature. The addition of Y to

an alloy improves the mechanical properties; this is primarily attributed to the Y-containing phases and fine microstructures [13,14]. These Y-containing phases have been shown to exhibit good heat resistance, with the capacity to improve high-temperature mechanical properties [7,15]. Liu et al [7] have reported that as-aged Mg-10Gd-3Y-1.2Zn-0.5Zr (wt.%) alloy demonstrated high-temperature strength because of $Mg_3$ (Gd,Y), $Mg_{24}Y_5$ and LPSO (long period stacking ordered) phases.

Our previous investigations have developed new Mg-6Zn-1Mn-4Sn-(0.1, 0.5 and 1.0) Y high strength wrought alloys which contain $Mg_2Sn$ and MgSnY high-temperature phases and show superior performance at room temperature [16]. However, the high-temperature tensile properties of Mg-Zn-Mn-Sn-Y alloys have not been reported. Therefore, this paper focusses on the high temperature tensile properties and fracture mechanism of as-aged Mg-6Zn-1Mn-4Sn-(0.1, 0.5 and 1.0) Y alloys.

## 2. Experimental Procedure

The experimental alloys were prepared with pure Mg (>99.9%, mass fraction), pure Zn (>99.95%), pure Sn (>99.9%), and Mg-30.0%Y and Mg-5.1%Mn master alloys melted at about 750 °C in a ZG-0.01 vacuum induction melting furnace in an Ar atmosphere. The chemical compositions were analyzed by XRF-18005 CCDE (Shimadzu, Kyoto, Japan) sequential X-ray fluorescence spectrometer; the results are presented in Table 1. The dimensions of samples were: 25 mm in diameter and 5 mm in height. The ingots were homogenized at 420 °C for 12 h, and then extruded into bars of 16 mm in diameter at 360 °C with an extrusion ratio of 25. After extrusion, the alloys were a solid solution at 440 °C for 2 h, then two-step aged at 90 °C for 24 h and 180 °C for 8 h. High-temperature tensile tests were carried out by a SANS CMT-5105 (MTS, Shenzhen, China) electronic testing machine at temperatures of 150 °C, 200 °C, 250 °C, and 300 °C. The tensile samples were machined in accordance with the GB/T4338-1995 standard [17]. The tensile tests were performed at a constant strain rate of $10^{-3}$ s$^{-1}$ with the load direction parallel to the specimen axis, and the yield strength (YS), ultimate tensile strength (UTS), and elongation were averaged from three samples. The microstructures were examined by a LEXT 4000 (Olympus, Tokyo, Japan) optical microscope (OM), a Rigaku D/max 2500PC (Riguka, Tokyo, Japan) X-ray diffractometer using Cu K$\alpha$ (XRD), an ESCAN VEGA (VEGA 3 XMU, ESCAN, BrNo, Czech Republic) scanning election microscope (SEM), and a JEM 2010 (200 kV, JEM, Tokyo, Japan) transmission electrical 25 microscope (TEM).

**Table 1.** Actual compositions of Mg-6Zn-1Mn-4Sn-Y alloys.

| Alloys | Actual Compositions (wt.%) | | | | |
|---|---|---|---|---|---|
| | **Mg** | **Zn** | **Mn** | **Sn** | **Y** |
| Mg-6Zn-1Mn-4Sn-0.1Y (ZMT614-0.1Y) | 88.57 | 6.09 | 0.99 | 4.22 | 0.13 |
| Mg-6Zn-1Mn-4Sn-0.5Y (ZMT614-0.5Y) | 88.07 | 6.14 | 0.91 | 4.38 | 0.50 |
| Mg-6Zn-1Mn-4Sn-1.0Y (ZMT614-1.0Y) | 88.07 | 6.10 | 0.98 | 3.91 | 0.94 |

## 3. Results and Discussion

### 3.1. As-Aged (Two Step-Aged) Microstructures

Figure 1 shows the XRD patterns of as-aged ZMT614-Y alloys. The phase compositions of the samples are $\alpha$-Mg, $\alpha$-Mn, $MgZn_2$, $Mg_2Sn$, and MgSnY phases. The $Mg_2Sn$ and MgSnY phases, for which phase transition temperatures are over 460 °C, provide excellent heat resistant performance [18,19]. For the Mg-4Zn-4.4Sn-0.6Y alloys, the MgSnY phase first forms during the solidification, and does not dissolve or decompose in the solid solution process (470 °C/(2 h)) [18,20]. As a result, the MgSnY phase cannot precipitate during the aging process.

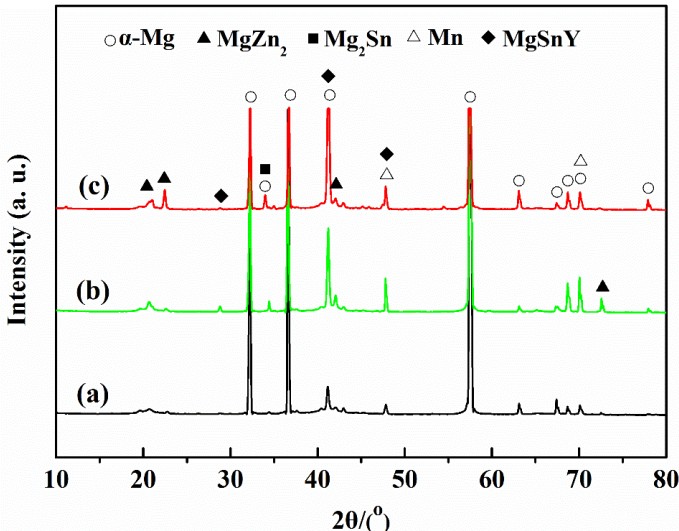

**Figure 1.** XRD patterns of as-aged ZMT614-Y alloys. (**a**) ZMT614-0.1Y, (**b**) ZMT614-0.5Y, (**c**) ZMT614-1.0Y.

Figure 2a–c shows the optical microstructures of the as-aged ZMT614-0.1Y, ZMT614-0.5Y, and ZMT614-1.0Y samples, and Figure 2d shows the TEM microstructures of as-aged ZMT614-0.5Y alloy. The grains were refined, and as the volume fraction of the second phase increased, the Y content increased. Tiny $Mg_2Sn$ and coarse MgSnY particles distribute in the grains and boundaries. The grain sizes decreased monotonously as the Y content increased. The average grain sizes of ZMT614-0.1Y, ZMT614-0.5Y, and the ZMT614-1.0Y alloy were (47.6 ± 1.5) μm, (45.3 ± 2.1) μm, and (42.5 ± 1.8) μm, respectively, indicating that Y and its compounds inhibit grain growth in the heat treatment process. As shown in Figure 2d, some blocky dark compounds and high-density gray nanoscale particles disperse in the matrix. Previous studies indicate that these coarse compounds are $Mg_2Sn$ and MgSnY phases, and the nanoscale particles are the $MgZn_2$ phase precipitated during the aging process [16,18].

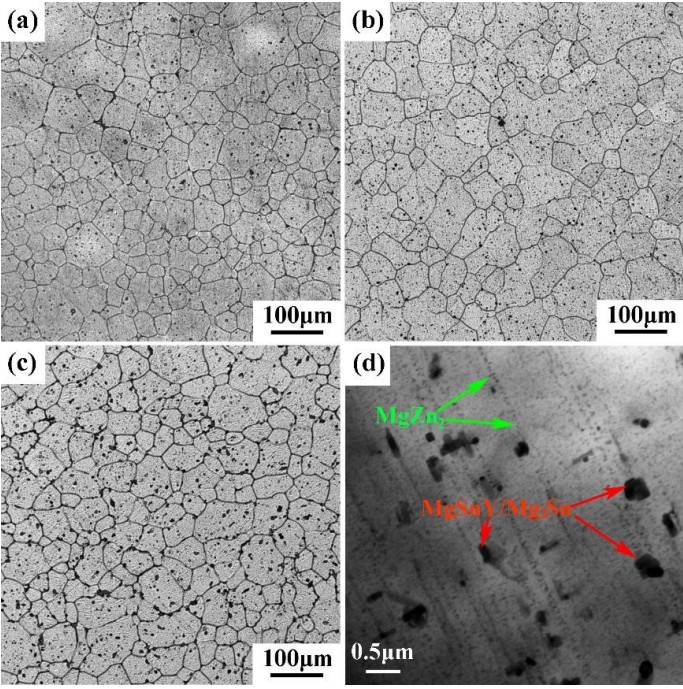

**Figure 2.** Microstructures of as-aged ZMT614-Y alloys. (**a**), (**b**), and (**c**) are the optical microstructures of ZMT614-0.1Y, ZMT614-0.5Y, and ZMT614-1.0Y, respectively; (**d**) is the TEM image of the ZMT614-0.5Y alloy.

### 3.2. High Temperature Tensile Properties

Figure 3 shows the stress-strain curves of as-aged ZMT614-1.0Y alloy tested at different temperatures. These curves were divided into four stages, namely, the work hardening stage, uniform deformation stage, localized necking stage, and fracture stage. At the initial stage of tension, the density of dislocations increases geometrically, which sharply increases the stress, and causes work hardening to occur. As the flow stress further increases, or even peaks, uniform deformation occurs on account of the dynamic balance between work hardening and dynamic recovery or recrystallization. After that, the flow stress decreases until fracture. The UTS and YS decrease noticeably as the tensile temperatures increase from 150 °C to 300 °C, while the elongation only increases slightly.

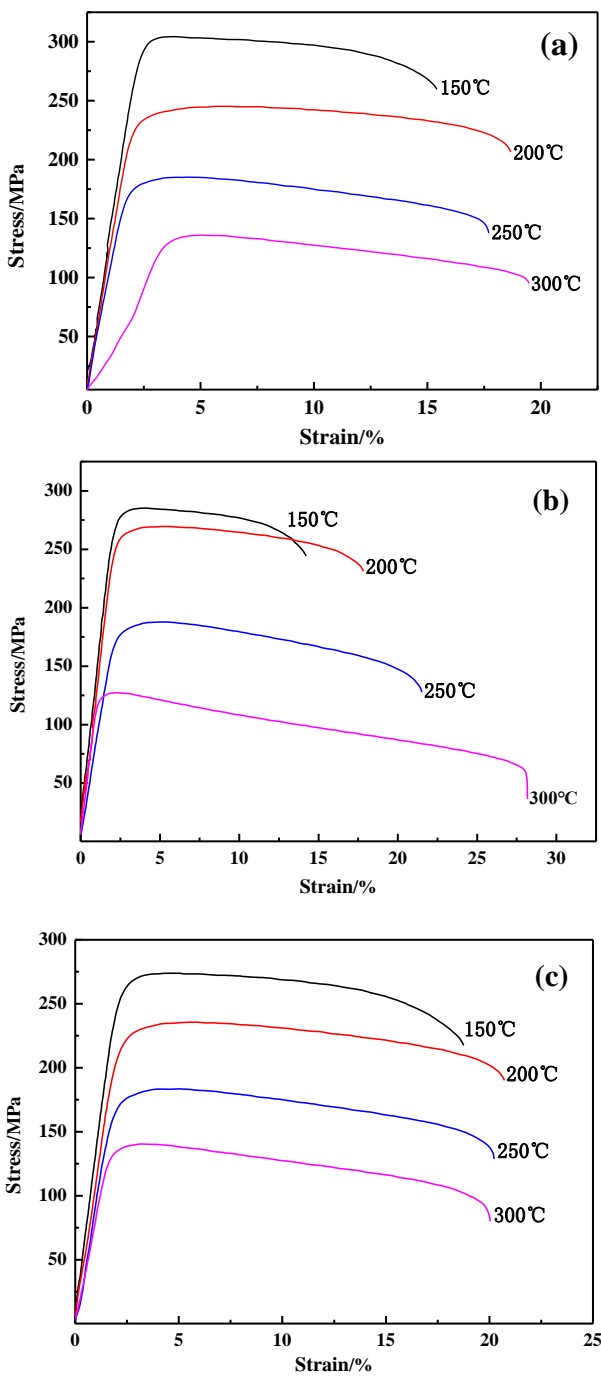

**Figure 3.** Stress-strain curves of T6-treated ZMT614-Y alloys tested at different temperatures. (**a**) ZMT614-0.1Y, (**b**) ZMT614-0.5Y, (**c**) ZMT614-1.0Y.

Figure 4 shows the corresponding high-temperature tensile mechanical properties. The ZMT614-0.5Y alloy exhibits the best mechanical properties. At 150 °C, the UTS, YS, and elongation values are 305 MPa, 277 MPa and 16.7%, respectively. When the tensile temperature increases to 300 °C, the UTS and YS decrease to 150 MPa and 136 MPa, respectively, and the elongation increases to 25.5%. The UTS, YS, and elongation of as-aged ZMT614-0.5Y alloy are 376 MPa, 371 MPa, and 7.7% at room temperature, respectively. As the tensile temperature ranges upwards from room temperature to 300 °C, the YS and UTS decrease by 226 MPa and 235 MPa, respectively, while the elongation increases by 17.5%. This suggests that the deformation mechanism fundamentally changes.

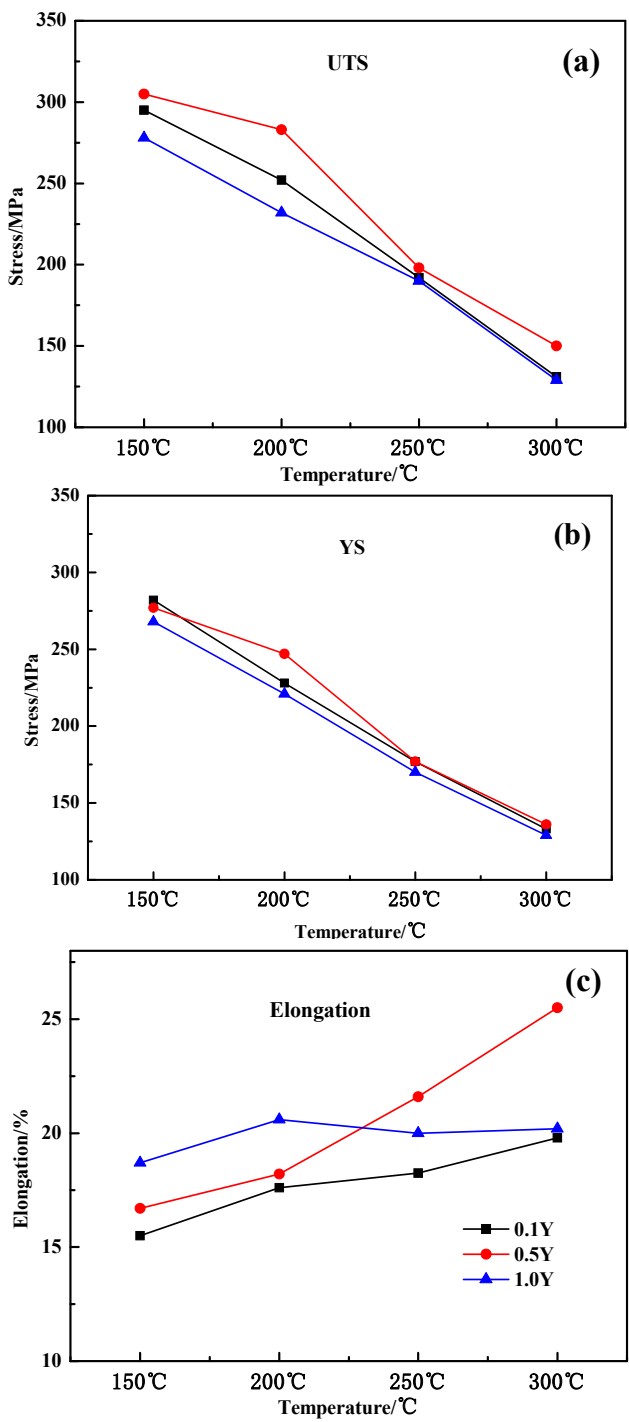

**Figure 4.** Tensile properties of T6-treated ZMT614-Y alloys tested at different temperatures. (**a**) UTS, (**b**) YS, (**c**) Elongation.

### 3.3. Fracture Mechanism

Figure 5 shows SEM and BSE images of the fracture morphology of as-aged ZMT614-1.0Y alloy tested at different temperatures. The fracture surfaces are covered with a large number of second phase particles and dimples, and have a high density of tearing edges and a small number of cavity defects. At a tensile temperature of 150 °C, several microscopic cavity defects and dimples exist in the fracture surface. As the tensile temperature increased to 200 °C, more microscopic cavity defects and blade-type edges were generated. When the tensile temperatures reached 250 °C and then 300 °C, the number of dimples decreased noticeably, and the number of cleavage planes with characteristic grain boundaries increased, showing characteristic intergranular fractures.

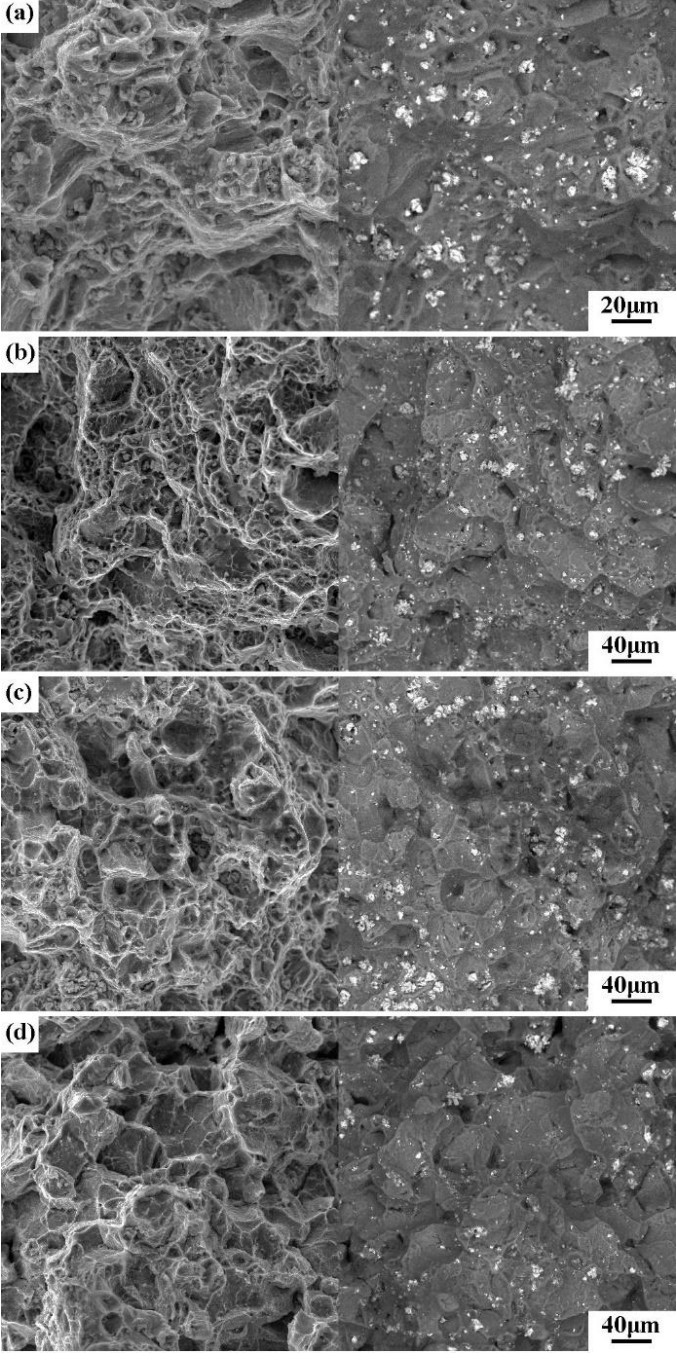

**Figure 5.** SEM (the left) and BSE (the right) images of the fracture morphology of T6-treated ZMT614-1.0Y alloys tested at different temperatures. (**a**) 150 °C, (**b**) 200 °C, (**c**) 250 °C, (**d**) 300 °C.

Figure 6 shows high-resolution SEM images of the fracture morphology of as-aged ZMT614-1.0Y alloy tested at different temperatures. The fracture mode is determined by the deformation temperature. As shown in Figure 6a,b, when the tensile temperature is 200 °C, the fracture surfaces will be covered with dimples and tearing edges, and exhibit transgranular fractures. When the tensile temperature reaches 250 °C, obvious grain boundaries sliding (GBS) operates (indicated by arrows in Figure 6c) at the bottom of big dimples. Higher deformation temperatures can promote the diffusion and migration of grain boundaries, facilitating the grain boundaries sliding. Consequently, the fracture surface exhibits a high density of cleavage planes as the deformation temperature increases to 300 °C.

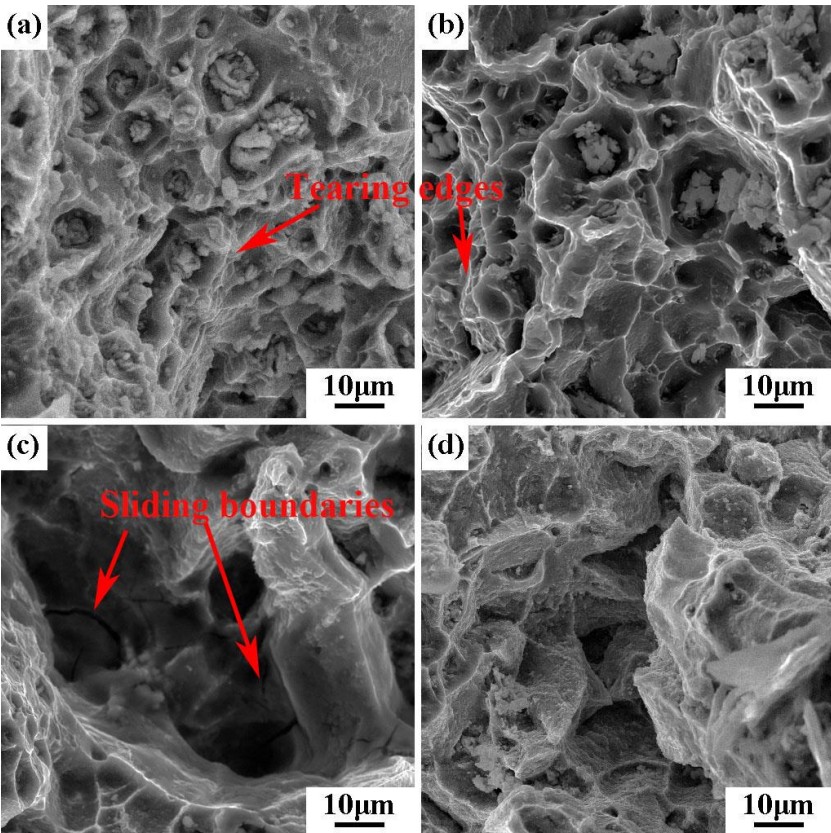

**Figure 6.** High resolution SEM images of fracture morphology of T6 treated ZMT614-1.0Y alloy tested at different temperatures. (**a**) 150 °C, (**b**) 200 °C, (**c**) 250 °C, (**d**) 300 °C.

Figure 7 shows the fracture morphology of a longitudinal section of as-aged ZMT614-1.0Y alloy tested at different temperatures. Sections (a), (c), (e), and (g) show the fracture surfaces of the longitudinal section, and (b), (d), (f), and (h) show the optical microstructures near the fracture surface.

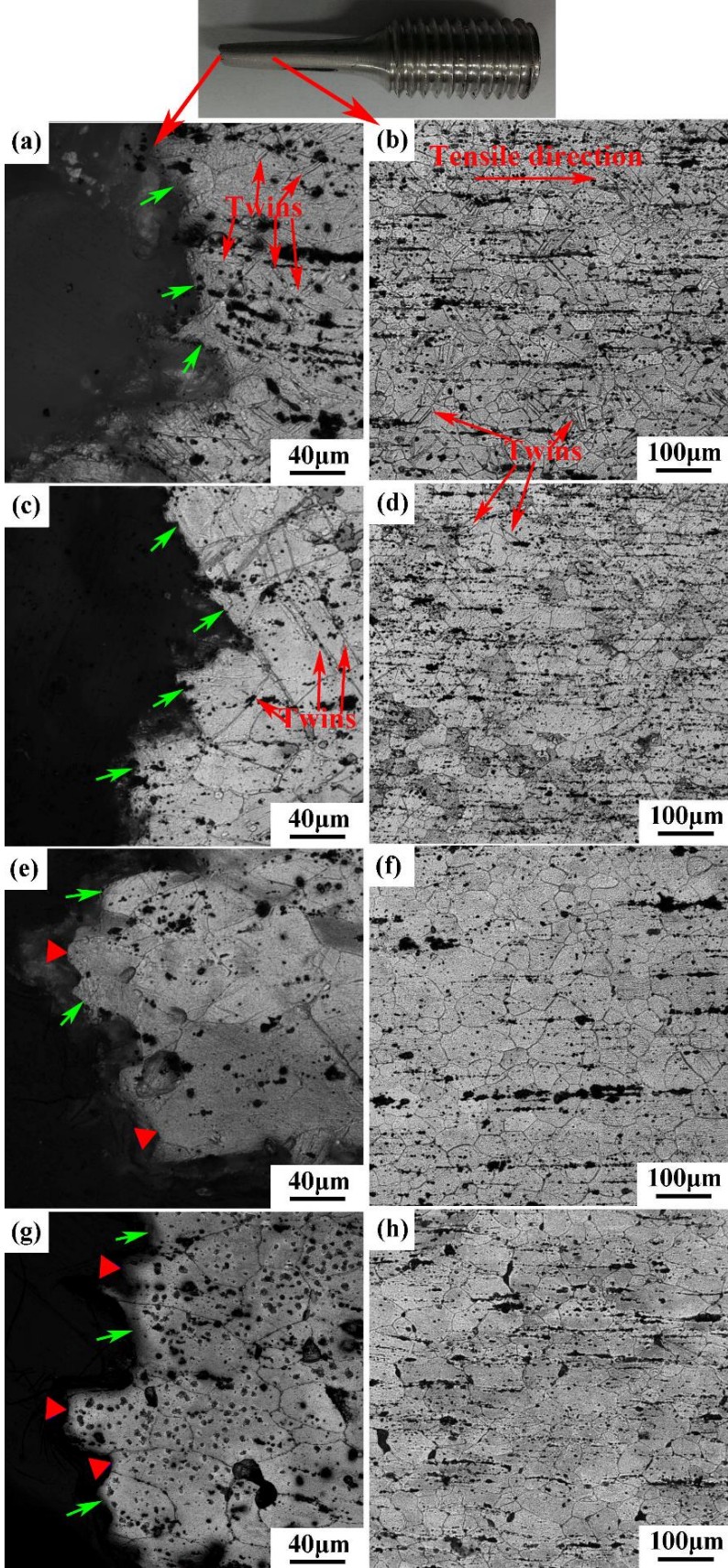

**Figure 7.** Fracture morphology of the longitudinal section of as-aged ZMT614-1.0Y alloy tested at different temperatures. (**a**), (**c**), (**e**), and (**g**) are the fracture surfaces of the longitudinal section, whereas (**b**), (**d**), (**f**), and (**h**) are the optical microstructures near the fracture surface.

To further consider the fracture mechanism, the longitudinal section microstructures of fracture surfaces of as-aged ZMT614-1.0Y alloys tested at different deformation temperatures are shown in Figure 7. There is a high density of twins dispersed in the fracture surface and matrix when tensile at 150 °C, revealing the main deformation mechanism to be basal plane slipping and twins. The density of twins decreases noticeably when the tensile temperature increases to 200 °C, which indicates that the main deformation mechanisms were basal plane, twins, and non-basal plane slipping. However, the twins disappear and voids form along the grain boundaries when the tensile temperature increases to 250 °C and then to 300 °C, suggesting non-basal plane systems were activated and GBS came into operation during the deformation. Additionally, the microstructural observation near the fracture surface shows that equiaxed grains were obtained when deformation temperatures were below 300 °C. At temperatures of 150 °C, 200 °C, and 250 °C, the average grain size is 43 µm, 46 µm, and 55 µm, respectively. However, at 300 °C, the grains near the fracture surface were elongated along the deformation direction and the average grain size is 63 µm. Compared with the as-aged average 10 grain size, grains grow considerably at 250 °C and 300 °C, revealing that GBS operates at these temperatures.

It is common knowledge that Mg alloys have a poor plastic deformation capacity at room temperature due to their limited independent slip systems in the hexagonal close-packed (HCP) structure. The critical resolved shear stress (CRSS) of a basal plane system is 0.5–0.7 MPa at room temperature, while the CRSS of non-basal plane slip systems is a hundred times higher [21]. Consequently, the principal slip system is the basal plane system at room temperature. When the deformation temperature increases, the CRSS of non-basal plane slip systems begins to decrease as the amplitude of atomic vibration increases. Some potential non-basal plane slip systems, such as prismatic and pyramidal plane, are thus operated by thermal activation. For instance, the CRSS of a prismatic plane slip system borders that of a basal plane system at 300 °C. The operation of non-basal plane slip systems reduces the number of dislocations, which mitigates against the formation of twins. The deformation temperatures have a predominant effect on the deformation mechanism of Mg alloys: basal plane slip systems are activated and twins coordinate deformation under 225 °C, and non-basal plane slip systems are activated and grain boundary diffusion and migration appear above 225 °C. As shown in Figure 7c–f, the twins disappear when the tensile temperature is over 200 °C, suggesting that non-basal plane systems are activated. From Figure 7a,c, it can be seen that the fracture surfaces are rough and irregular (indicated by the arrows). The fracture surfaces in Figure 7e,g are characteristically even and regular (indicated by the triangle). By combining all the analysis results, three types of fracture mechanisms can be distinguished. When tensile temperatures are 150 °C and 200 °C, the fracture mechanism is of the transgranular type. As the temperature increases to 250 °C, the fracture type is a mixed mode of transgranular and intergranular types. At 300 °C, the fracture mechanism becomes solely the intergranular fracture type.

Figure 8 shows the BSE images of fracture morphology of as-aged ZMT614-1.0Y alloy tested at 150 °C and 300 °C. The chemical compositions of Mg, Sn, and Y of the coarse particle in Figure 8a are 45.20%, 33.14%, and 10.13% (at.%), respectively, which are approved as $Mg_2Sn$ and MgSnY phases [16,21]. As shown in Figure 8a,b, a high density of twins and coarse $Mg_2Sn$ and MgSnY particles are distributed in the matrix. As tensile temperature increases to 300 °C, different dimension voids appear in the grains, especially minute voids in the grain boundaries. Depending on the Mg-Zn ternary phase diagram, the eutectic temperature ($T_E$) of the $MgZn_2$ phase is about 340 °C. At the deformation temperature of 300 °C, which is $0.93T_E$ of the $MgZn_2$ phase, the $MgZn_2$ precipitates soften and lubricate the grains and boundaries, which promote the initiation and propagation of cracks. The $MgZn_2$ phase precipitates preferentially in the grain boundaries in the Mg-Zn-Mn alloys [22]. Therefore, the cracks nucleate preferentially in the grain boundaries, and result in intergranular fracture at higher deformation temperatures.

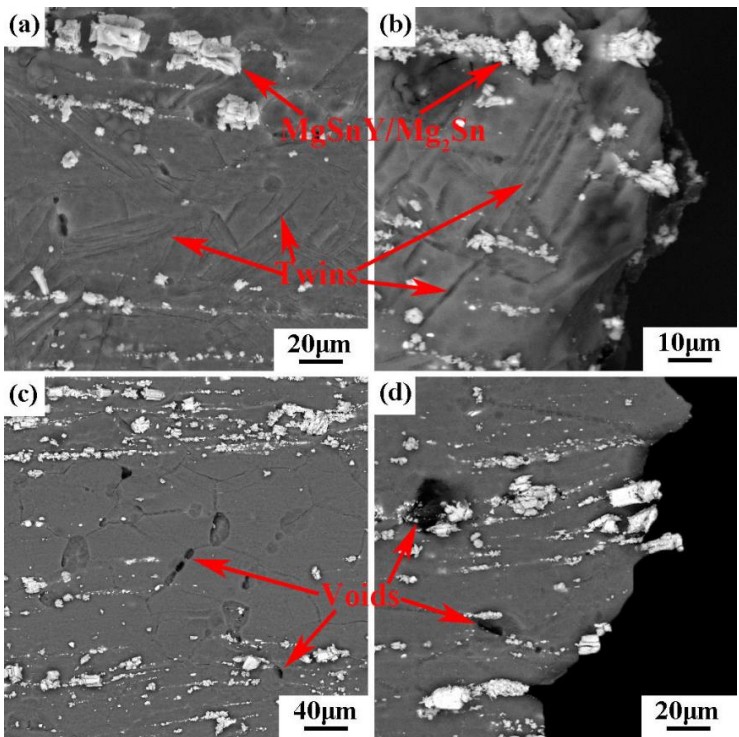

**Figure 8.** BSE fracture morphology of the longitudinal section of as-aged ZMT614-1.0Y alloys tensile at different temperatures: (**a**) and (**b**) at 150 °C, (**c**) and (**d**) at 300 °C.

## 4. Conclusions

The high-temperature mechanical properties and fracture mechanism of as-aged ZMT614-xY (x = 0.1, 0.5 and 1.0) alloys were investigated, and the following conclusions were drawn.

(1)　The phase compositions of as-aged ZMT614-Y alloys were $\alpha$-Mg, $\alpha$-Mn, $MgZn_2$, $Mg_2Sn$, and MgSnY phases. The $Mg_2Sn$ and MgSnY phases were high-temperature phases and inhibited grain growth in the high-temperature tensile process.

(2)　The ZMT614-0.5Y alloy exhibited the best high-temperature mechanical properties. At 150 °C, the UTS, YS, and elongation was 305 MPa, 277 MPa, and 16.7%, respectively. As the tensile temperature increased to 300 °C, the UTS, YS, and elongation decreased to 150 MPa, 136 MPa and 25.5%, respectively.

(3)　The tensile temperature exerted dominance over the fracture mechanism. At the tensile temperatures of 150 °C and 200 °C, the fracture mechanism was of the transgranular fracture type. At 250 °C, the fracture mechanism was of the transgranular and intergranular mixed mode fracture type. As the tensile temperature reached 300 °C, the fracture mechanism changed to be solely of the intergranular fracture type.

**Author Contributions:** Formal analysis, G.H. and M.Z.; Investigation, M.Z.; Methodology, G.H. and C.G.; Project administration, G.H.; Writing – original draft, G.H. and C.G.; Writing-review & editing, M.Z.

**Funding:** This research was supported by Zhejiang Provincial Natural Science Foundation of China under Grant No. LQ18E010003, No. LQ18E010004 and Jiaxing Science and Technology Plan Funded Project(2018AY11004).

**Conflicts of Interest:** The authors declare no conflict of interest.

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
