# Peer review of "Microstructures and High Temperature Tensile Properties of As-Aged Mg-6Zn-1Mn-4Sn-(01, 0.5 and 1.0) Y Alloys"

_metals, doi:10.3390/met9010001_

Round 1
Reviewer 1 Report
It is suggested the authors carefully reread the manuscript Example are as follows:
Title, key words and lines 46-47, 89 etc: “High temperature mechanical properties” only high temperature tensile properties are described in the manuscript.
Title & line 44: “Mg-Zn-Mn-Sn-Y Alloys” Since the authors investigated x=0.1, 0.5, 1.0, wt.% Y additions to Mg-6Zn-1Mn-4Sn-xY, perhaps a better title could change be: “Mg-6Zn-1Mn-4Sn-xY for x=0.1-1 wt.%....” (or similar).
Abstract: “yield strength (YS), ultimate tensile strength (UTS) and elongation was 305MPa, 277MPa and 16.7% at 150℃, respectively.” This is not in agreement with the manuscript text.
YS < UTS
“YS and UTS decreased to 150MPa and 136MPa”
Again, YS < UTS
Abstract: please mention the temperature range or temperatures at which the high temperature tensile properties were investigated.
Lines 18-22: cleavage or ductile fracture mechanisms?
Lines 35-37: please specify which RE elements…such as…
Line 41: it should read “showed”
Line 42: in wt.% throughout the manuscript.
Line 43: “Mg3 (Gd, Y)” No spaces”
Line 54: please specify the ingot size? and/or weight?
Line 57: “under” change to “at”.
Line 59: “with GB/T4338-1995 standard [Reference(s)?]
“1*10-3s-1” please omit “1*”
Table 1: Please replace Mg values by “balance”
Line 68: please confirm here whether the microstructures were examined after the two-step aged condition.
Line 71: “For the Mg-Zn-Mn-Sn-Y alloys,”…please specify the composition range.
Line 77: “Figure 2 shows optical and TEM microstructures of the as- aged samples.” Please re-write, not all of the TEM samples are shown here.
Line 77-78: “The grains get refined and the volume fraction of second phase increases as the Y content increases.” This cannot be quite distinguished from just Figure 2.
Line 81: please give standard deviations.
Lines 83-85: “Based on previous studies, the coarse compounds are Mg2Sn and MgSnY phases, and the nanoscale particles are MgZn2 phase precipitated during the aging process [17, 18].” The authors could have experimentally confirm that.
Please enlarge all figures.
Line 91: “at different temperatures”. Please specify.
Figure 3: YS at 0.2%? etc.
Figure 4: Please use the same Y-axis scale for strength in Fig. 4(a)&(b).
Line 114 and Figure 5: “SEM and BSE images…”? BSE images are SEM images.
Figure 5 caption: “ZMT614-Y alloys” what alloys correspond the images in Figure 5
Lines 119-120: “When the tensile temperatures reach 250℃ and 300℃,
the amount of dimples decreases obviously.” descreases or increases?
Author Response
RESPONSES TO REVIEWERS’ COMMENTS
Thank you for your useful comments and suggestions our manuscript. We have modified the manuscript accordingly, and detailed corrections are listed below point by point, and the revision is highlighted in red. We have our manuscript checked by a native English speaking colleague, and the revision is highlighted in blue.
REVIEWER 1#:
1) In Title, key words and lines 46-47, 89 etc: “High temperature mechanical properties” only high temperature tensile properties are described in the manuscript. Title & line 44: “Mg-Zn-Mn-Sn-Y Alloys” Since the authors investigated x=0.1, 0.5, 1.0, wt.% Y additions to Mg-6Zn-1Mn-4Sn-xY, perhaps a better title could change be: “Mg-6Zn-1Mn-4Sn-xY for x=0.1-1 wt.%....” (or similar).
According to the reviewer’s suggestion, we have revised the title as “Microstructures and High Temperature Tensile Properties of As-Aged Mg-6Zn-1Mn-4Sn-xY Alloys”. The keywords and lines 44, 46-47, 89 etc. have been revised.
2) In Abstract: “yield strength (YS), ultimate tensile strength (UTS) and elongation was 305MPa, 277MPa and 16.7% at 150℃, respectively.” This is not in agreement with the manuscript text. YS < UTS“YS and UTS decreased to 150MPa and 136MPa” Again, YS < UTS.
According to the reviewer’s suggestion, we have revised the value of YS and UTS.
3) Abstract: please mention the temperature range or temperatures at which the high temperature tensile properties were investigated.
According to the reviewer’s suggestion, we have added the “The tensile temperatures were 150℃, 200℃, 250℃ and 300℃, respectively.” into the abstract.
4) Lines 18-22: cleavage or ductile fracture mechanisms?
The fracture mechanism is ductile fracture.
5) Lines 35-37: please specify which RE elements…such as….
According to the reviewer’s suggestion, we have added the “such as Nd, Ce and La” into the text.
6) Line 41: it should read “showed”.
According to the reviewer’s suggestion, we have revised the “show” as “showed” in the text.
7) Line 42: in wt.% throughout the manuscript.
According to the reviewer’s suggestion, we have added the “(wt.%)” into the text.
8) Line 43: “Mg3 (Gd, Y)” No spaces”.
I’m very sorry to make such a mistake. According to the reviewer’s suggestion, we have eliminated the space.
9) Line 54: please specify the ingot size? and/or weight?
According to the reviewer’s suggestion, we have specified the ingot size in the text.
10) Line 57: “under” change to “at”.
According to the reviewer’s suggestion, we have changed the “under” to “at” in the text.
11) Line 59: “with GB/T4338-1995 standard [Reference(s)?].
According to the reviewer, we have added the GB/T4338-1995 standard as reference 18 in the text.
12) “1*10-3s-1” please omit “1*”.
According to the reviewer’s suggestion, we have omit “1*” in the text.
13) Table 1: Please replace Mg values by “balance”.
We have replace Mg values by “balance” in the text.
14) Line 68: please confirm here whether the microstructures were examined after the two-step aged condition.
We have confirmed the microstructures after the two-step aged condition were examined in the text.
15) Line 71: “For the Mg-Zn-Mn-Sn-Y alloys,”…please specify the composition range.
The composition range of the alloys have been specified in the text.
16) Line 77: “Figure 2 shows optical and TEM microstructures of the as-aged samples.” Please re-write, not all of the TEM samples are shown here.
In consideration of the comments of the reviewer, we have re-write the Figure 2 as “Figure 2a, 2b and 2c show the optical microstructures of the as-aged samples, and Figure 2d shows the TEM microstructures of as-aged ZMT614-0.5Y alloy.” in the text.
17) Line 77-78: “The grains get refined and the volume fraction of second phase increases as the Y content increases.” This cannot be quite distinguished from just Figure 2.
As shown in the Figure 2a, 2b and 2c, the amount of dark particles increases as the Y content increase. The dark particles are the second phases. And in the below, the average grain size decreases as the Y content increases.
18) Line 81: please give standard deviations.
The standard deviations of grain size are given in the text.
19) Lines 83-85: “Based on previous studies, the coarse compounds are Mg2Sn and MgSnY phases, and the nanoscale particles are MgZn2 phase precipitated during the aging process [17, 18].” The authors could have experimentally confirm that. Please enlarge all figures
These phases have been confirmed in our previous experiment and have been showed in our published papers.
20) Line 91: “at different temperatures”. Please specify.
We have specified the different temperatures in the text.
21) Figure 3: YS at 0.2%? etc.
The YS is at 0.2% of the curves.
22) Figure 4: Please use the same Y-axis scale for strength in Fig. 4(a)&(b).
We have revised the Fig. 4(a)&(b) by using the same Y-axis scale and added them into the text.
23) Line 114 and Figure 5: “SEM and BSE images…”? BSE images are SEM images.
The BSE images are not SEM images. The BSE images are back scattered electron images. These images could distinguish the second phases by color contrast.
24) Figure 5 caption: “ZMT614-Y alloys” what alloys correspond the images in Figure 5.
I’m so sorry to make such a mistake. The “ZMT614-Y alloys” is the “ZMT614-1.0Y alloys” and we have revised in the text.
25) Lines 119-120: “When the tensile temperatures reach 250℃ and 300℃, the amount of dimples decreases obviously.”decreases or increases?
Yes, the amount of dimples decreases as shown in the Figure 5. The cleavage planes with grain boundaries character and cavity defect increase.

Reviewer 2 Report
The paper deals with the microstructure and high temperature mechanical properties of three as-aged Mg-Zn-Mn-Sn-Y alloys, having different yttrium contents. The paper is of potential interest for the readers, but remarkable improvements must be implemented, in order for it to be considered for publication.
Major Comments:
-There is a lot of confusion throughout the text concerning the labels of the alloys developed, the authors must check carefully.
-Statements concerning the phases visible in TEM micrographs must be confirmed with SAED investigations. Also, why the authors show TEM results for one alloy composition only?
-From Figure 5, all the investigations refer only to the 1-.0Y alloy composition, why?
There is a lot of work missing on microstructural characterization concerning the other two alloy formulations.
-Figure 1 must be improved, please also label each pattern with the corresponding sample.
-Figure 3 and Figure 4 have poor quality and must be improved.
-The annotations in figure 7 are very hard to read, please change colors and improve them.
-In the discussion of Figure 8, authors make statements concerning the composition of the particles in the matrix, but they do not bring any evidence to support their statements. EDS data are needed. Also, the authors must state if micrographs are taken with SE or BSE signal.
-There are typos throughout the text and the captions, please check carefully.
Author Response
RESPONSES TO REVIEWERS’ COMMENTS
Thank you for your useful comments and suggestions our manuscript. We have modified the manuscript accordingly, and detailed corrections are listed below point by point, and the revision is highlighted in red. We have our manuscript checked by a native English speaking colleague, and the revision is highlighted in blue.
REVIEWER 2#:
1) There is a lot of confusion throughout the text concerning the labels of the alloys developed, the authors must check carefully.
I’m very sorry to make such mistakes. We have checked the text carefully and corrected these mistakes.
2) Statements concerning the phases visible in TEM micrographs must be confirmed with SAED investigations. Also, why the authors show TEM results for one alloy composition only?
These SAED investigations have been showed in our pervious published papers and cited in this paper. Due to the types of the second phase of these alloys are the same, we take one alloy as representative.
3) From Figure 5, all the investigations refer only to the 1.0Y alloy composition, why? There is a lot of work missing on microstructural characterization concerning the other two alloy formulations.
In our experiments, all the alloys are tested. However, the fracture morphology of these alloys show similar. If added all the figures into this paper, to many figures would be repetitious and take a lot of space. In order to avoid repetition, we take the ZMT614-1.0 alloy as representative.
4) Figure 1 must be improved, please also label each pattern with the corresponding sample.
We have improved the Figure 1 and added it into the text.
5) Figure 3 and Figure 4 have poor quality and must be improved.
We have added high quality figures in the text.
6) The annotations in figure 7 are very hard to read, please change colors and improve them.
We have changed the annotations color in the Figure 7 and added into the text.
7) In the discussion of Figure 8, authors make statements concerning the composition of the particles in the matrix, but they do not bring any evidence to support their statements. EDS data are needed. Also, the authors must state if micrographs are taken with SE or BSE signal.
According to the reviewer’s suggestion, we have added the EDS results in the text. The micrographs are taken with BSE signal.
8) There are typos throughout the text and the captions, please check carefully.
I’m so sorry to make such mistakes. We have check the text carefully and revised these typos.

Reviewer 3 Report
The authors made a nice study of the microstructures and mechanical behavior at high temperature of Mg-6Zn- 1Mn-4Sn alloys by using non-destructive (basically, microscopy) and destructive (tensile tests) methods. The paper is well written and structured and the results support the conclusions, so it should be accepted if some minors considerations are taken account:
1) Section 2). Please, include some logical diagram and pictures of the experimental procedure and setup.
2) In general, quality of the figures is very poor. It must be improved.
3) Figure 2d. Micrography is so dark. Please, try to improve it.
4) Figure 3. Please, what do you mean with “Engineering strian”?. Please, use strain
Author Response
RESPONSES TO REVIEWERS’ COMMENTS
Thank you for your useful comments and suggestions our manuscript. We have modified the manuscript accordingly, and detailed corrections are listed below point by point, and the revision is highlighted in red. We have our manuscript checked by a native English speaking colleague, and the revision is highlighted in blue.
REVIEWER 3#:
1) Section 2). Please, include some logical diagram and pictures of the experimental procedure and setup.
The logical diagram and pictures of the experimental procedure and setup are normal. These are not beneficial to the text. And these equipment models are shown in the paper.
2) In general, quality of the figures is very poor. It must be improved.
We have added high quality figures in the text.
3) Figure 2d. Micrography is so dark. Please, try to improve it.
The improved Figure 2d have been added in the text.
4) Figure 3. Please, what do you mean with “Engineering strian”? Please, use strain
We have used the strain in the Figure 3 and added it into the text.

Round 2
Reviewer 1 Report
The authors have revised the paper according to my suggestions and have addressed most them. However, there is still divergence of opinion on the main points below.
Perhaps the title should read: Mg-6Zn-1Mn-4Sn-(0.1, 0.5 and 1)Y (optional suggestion)
Line 10: “x=0.1, 0.5, 1.0” change to “x=0.1, 0.5 and 1.0”
Lines 19 – 23: is that for all the 3 composition investigated?
Lines 37 – 38: “rare earth (RE)” should have been defined before.
Lines 47 – 51: xY please define x.
Table 1: it should read: Mg bal.
Fig. 2 caption. ZMT614-0.1Y, (b) ZMT614-0.5Y and (c) ZMT614-1.0Y
Fig. 4: Please give the 3 values or average ± std. dev. bars as overlapping can occur.
Figs. 6 and 8 are the same.
Fig. 7 caption: please specify the temperatures.
Lines 142 – 143: It is a bit odd to have more cleavage planes at the highest testing temperature of 300oC, unless there is embrittlement. Could the authors explain this a bit better.
Lines 186 – 187: “rough and irregular”, “even and regular”. Those terms are relative. The authors should either mark them in the corresponding figures better or omit those terms.
References: please omit [J].
Please
enlarge the figures. Page 13 is almost blank.
Author Response
RESPONSES TO REVIEWERS’ COMMENTS
Thank you for your useful comments and suggestions our manuscript. We have modified the manuscript accordingly, and detailed corrections are listed below point by point. The revision is highlighted in red.
REVIEWER 1#:
1) Perhaps the title should read: Mg-6Zn-1Mn-4Sn-(0.1, 0.5 and 1)Y (optional suggestion).
According to the reviewer’s suggestion, we have revised the title as “Microstructures and High Temperature Tensile Properties of As-Aged Mg-6Zn-1Mn-4Sn-(0.1, 0.5 and 1.0)Y Alloys”.
2) Line 10: “x=0.1, 0.5, 1.0” change to “x=0.1, 0.5 and 1.0”.
According to the reviewer’s suggestion, we have revised the mistakes.
3) Lines 19-23: is that for all the 3 composition investigated?
Yes, we have investigated all the 3 compositions.
4) Lines 37-38: “rare earth (RE)” should have been defined before.
According to the reviewer’s suggestion, we have defined the rare earth in the line 37.
5) Lines 47-51: xY please define x.
According to the reviewer’s suggestion, we have defined the xY as x in the text.
6) Table 1: it should read: Mg bal.
Based on the two reviewer’s suggestions, we have revised the "Mg" in the text.
7) Fig. 2 caption. ZMT614-0.1Y, (b) ZMT614-0.5Y and (c) ZMT614-1.0Y.
According to the reviewer’s suggestion, we have revised the Figure 2 caption the text.
8) Fig. 4: Please give the 3 values or average ± std. dev. bars as overlapping can occur.
The values in the text are the average values already. The YS and UTS of these alloys are a little differences. As a result, we research emphases on the variation tendency of tensile properties and fracture modes of these alloys in this paper.
9) Figs. 6 and 8 are the same.
I’m so sorry to make such mistakes. We have revised the Figure 6 in the text.
10) Lines 142-143: It is a bit odd to have more cleavage planes at the highest testing temperature of 300oC, unless there is embrittlement. Could the authors explain this a bit better.
When the tensile temperature reaches to 300℃, the grain boundaries become brittle and the fracture mode is intergranular fracture. As a result, more cleavage planes appear.
11) Lines 186-187: “rough and irregular”, “even and regular”. Those terms are relative. The authors should either mark them in the corresponding figures better or omit those terms.
We have marked them in the Figure 7a, 7c, 7e and 7f.
12) References: please omit [J].
We have removed all the [J] in the references.
13) Please enlarge the figures. Page 13 is almost blank.
We have enlarge the figures in the text.
Reviewer 2 Report
The authors addressed properly all the comments, and now the quality of the paper is strongly improved. Some minor comments that should be considered are listed as follows:
-Please change the style of the References according to the Journal's Guide for authors;
-In Table 1, Please type Mg instead of "Bal." - The values corresponding to th emagnesium content (88.57 etc.) should be substituted with the word"Bal."
-Figure 1: please use different colors for each XRD pattern
-Title 3.2 (Line 94): please type "High temperature tensile properties"
-Line 57: the sentence "The samples were φ25 × 5 mm" is not clear, please change it.
-Figure 5 caption: please type "SE and BSE SEM micrographs...".
Author Response
RESPONSES TO REVIEWERS’ COMMENTS
Thank you for your useful comments and suggestions our manuscript. We have modified the manuscript accordingly, and detailed corrections are listed below point by point. The revision is highlighted in red.
REVIEWER 2#:
1) Please change the style of the References according to the Journal's Guide for authors.
We have revised the style of the references in the text.
2) In Table 1, Please type Mg instead of "Bal." - The values corresponding to the magnesium content (88.57 etc.) should be substituted with the word "Bal.".
Based on the two reviewer’s suggestions, we have revised the "Mg" in the text.
3) Figure 1: please use different colors for each XRD pattern.
We have added different colors XRD pattern into the text.
4) Title 3.2 (Line 94): please type "High temperature tensile properties".
We have revised the title 3.2 as "High temperature tensile properties" in the text.
5) Line 57: the sentence "The samples were φ25 × 5 mm" is not clear, please change it.
We have revised the sentence as "The dimension of samples were 25mm in diameter and 5 mm in height" and added into the text.
6) Figure 5 caption: please type "SE and BSE SEM micrographs...".
We have added the type of SE and BSE SEM micrographs in the Figure 5 caption.